# Eggshell Membrane Ameliorates Hyperuricemia by Increasing Urate Excretion in Potassium Oxonate-Injected Rats

**DOI:** 10.3390/nu13103323

**Published:** 2021-09-23

**Authors:** Yoon-Young Sung, Dong-Seon Kim

**Affiliations:** KM Science Research Division, Korea Institute of Oriental Medicine, 1672 Yuseong-daero, Yuseong-gu, Daejeon 34054, Korea; yysung@kiom.re.kr

**Keywords:** ATP-binding cassette transporter G2, gout, urate transporter 1, uric acid, potassium oxonate

## Abstract

Hyperuricemia is the primary cause of gouty arthritis and other metabolic disorders. Eggshell membrane (EM) is an effective and safe supplement for curing pain and stiffness connected with osteoarthritis. However, the effect of EM on hyperuricemia is unclear. This study determines the effects of EM on potassium oxonate-injected hyperuricemia. Uric acid, creatinine, blood urea nitrogen concentrations in the serum, and xanthine oxidase activity in the liver are measured. Protein levels of renal urate transporter 1 (URAT1), organic anion transporters 1 (OAT1), glucose transporter 9 (GLUT9), and ATP-binding cassette transporter G2 (ABCG2) in the kidney are determined with renal histopathology. The results demonstrate that EM reduces serum uric acid levels and increases urine uric acid levels in hyperuricemic rats. Moreover, EM downregulates renal URAT1 protein expression, upregulates OAT1 and ABCG2, but does not change GLUT9 expression. Additionally, EM does not change xanthine oxidase activity in the liver or the serum. EM also decreases uric acid uptake into oocytes expressing hURAT1. Finally, EM markedly reduces renal inflammation and serum interleukin-1β levels. These findings suggest that EM exhibits antihyperuricemic effects by promoting renal urate excretion and regulating renal urate transporters. Therefore, EM may be useful in the prevention and treatment of gout and hyperuricemia.

## 1. Introduction

Eggshell membrane (EM) is the film attached to the inside of the eggshell and is one of the components of eggs, a major food for human health. It allows for the penetration of oxygen and blocks the invasion of microorganisms. EM is composed of organic material (70%), nonorganic material (10%), and water (20%), with 80% of the organic material made up of protein. In addition, it is composed of 2.3% fat and 3.4% carbohydrates [1,2]. Among the components of eggs, egg yolk and egg whites are used as raw materials for food, and eggshells are a raw material for calcium supplements to form and maintain teeth and bones [3]. However, EM is classified as a waste and is not frequently utilized.

The anti-arthritic effects of EM have been recently reported in a lipopolysaccharide-induced animal model and in human clinical studies [4,5]. EM has also improved inflammation in rats with collagen-induced rheumatoid arthritis [6]. In addition, EM has demonstrated anti-arthritic activity in monosodium iodoacetate-induced osteoarthritis rats [7]. Monosodium iodoacetate is an inducer of gout arthritis, and the accumulation of monosodium urate crystals in the joints can result in inflammation, leading to gouty arthritis. Hyperuricemia can cause crystals of uric acid (or urate) to form, and the deposition of these crystals in the joints causes gout, a form of arthritis that can be very painful. EM is a source of collagen, chondroitin sulfate, glucosamine, hyaluronic acid, and calcium, which have all been extensively investigated for the treatment of osteoarthritis [8]. The results of these studies show the potential usefulness of EM to prevent and treat gout and hyperuricemia.

Hyperuricemia is a major risk factor for the progress of insulin resistance, diabetes, obesity, hypertension, atherosclerosis, and cardiovascular disease, as well as gouty arthritis [9,10,11]. The appropriate control of hyperuricemia contributes to the prevention and treatment of these diseases. Hyperuricemia occurs with increased uric acid production or impaired uric acid excretion [12]. By reducing the production and enhancing the excretion of uric acid, urate-lowering treatment can play a crucial role in the control of hyperuricemia and hyperuricemia-associated disorders [13]. Commonly used urate-lowering medicines such as allopurinol and febuxostat, an inhibitor of uric acid synthesis, are widely used for the care of gout; however, these xanthine oxidase (XO) inhibitors can have severe adverse reactions, such as allopurinol hypersensitivity syndrome [14,15,16]. Benzbromarone is a uricosuric medication that has been used to control gout for the last 30 years, but due to severe hepatotoxicity side effects, it was withdrawn from the European market [17]. Thus, there is a need to develop drugs derived from natural products with no toxicity that are more effective for the prevention and long-term treatment of hyperuricemia-associated disorders. Therefore, this study investigates whether EM protects against hyperuricemia.

This study assesses the antihyperuricemic activity of EM in rats with hyperuricemia induced by potassium oxonate (PO). PO is a selectively competitive uricase inhibitor, which is widely used to inhibit the effect of liver uricase, which leads to hyperuricemia [18,19]. Thus, PO-treated rats are a practical animal model not only to investigate the pathology of hyperuricemia but also to evaluate potential medications [20]. To examine the activity and underlying mechanisms of EM on hyperuricemia, the dual mechanisms of EM via the inhibition of uric acid production and the enhancement of uric acid excretion are examined in hyperuricemic animals.

## 2. Materials and Methods

### 2.1. Animals

Seven-week-old male Sprague Dawley rats were purchased from Orient Bio (Seongnam, Korea). They were adapted in an air-conditioned animal room at 22 ± 1 °C with 50% ± 10% humidity. The rats were given access to a standard diet and water ad libitum. The animal studies were approved by the Institutional Animal Care and Use Committee of the Korea Institute of Oriental Medicine, and the experiments were carried out in accordance with the committee guidelines (approval code. 19-024).

### 2.2. Hyperuricemia Induction and EM Administration

The EM used in this study was provided as a dried powder by JU-HWAN BIOCELL Co., Ltd. (Cheonan, Korea). To induce hyperuricemia, 150 mg/kg PO (a uricase inhibitor) was injected intraperitoneally into the animals. The following positive control drugs were used: allopurinol (Sigma, St. Louis, MO, USA) and benzbromarone (TCI, Tokyo, Japan). Before injection, the PO was dissolved in 0.5% carboxymethyl cellulose (CMC, Sigma, USA) and 0.1 M sodium acetate (Sigma, USA). To examine the antihyperuricemic activities of EM (Experiment 1, preliminary acute study), the rats were divided into five groups (5 animals per group): a normal group; a PO-injected hyperuricemia group (PO); a PO + 10 mg/kg allopurinol-administrated group; a PO + 100 mg/kg EM-administrated group; and a PO + 200 mg/kg EM-administrated group. In Experiment 1, samples were dissolved in CMC and administered orally (once) to the rats 1 h after PO injection for 1 day.

To investigate the dose-dependent activities and molecular mechanisms of EM (Experiment 2), the rats were divided into 7 groups (n = 5 each): a normal group (N); a PO-injected hyperuricemia group; a PO + 10 mg/kg allopurinol-administrated group; a PO + 50 mg/kg benzbromarone-administrated group; a PO + 50 mg/kg EM-administrated group; a PO + 100 mg/kg EM-administrated group; and a PO + 200 mg/kg EM-administrated group. Samples were dissolved in 0.5% CMC solution and administered orally (every day) to the rats 1 h after PO injection for 5 days.

### 2.3. Collection of Blood, Urine, and Tissue Samples

Blood was collected 2 h following drug administration, and urine was also collected after 2 h, using a metabolic cage following sample treatment. After the collection of the blood and urine, tissue (kidney and liver) samples were carefully divided and preserved at −80 °C for further assays. The serum was separated by centrifugation (2000× *g*, 15 min, 4 °C). The serum and urine samples were then used for biochemical analysis.

### 2.4. Analysis of Uric Acid Concentrations in Serum and Urine

The uric acid concentrations from the serum and urine were determined using a commercial uric acid assay kit (Biovision, Milpitas, CA, USA) according to instructions provided by the manufacturer.

### 2.5. Measurement of Proinflammatory Cytokines Levels in Serum

Serum interleukin (IL)-1β concentrations were measured using a rat IL-1β enzyme-linked immunosorbent assay (ELISA) kit (R&D Systems, Minneapolis, MN, USA).

### 2.6. In Vitro Xanthine Oxidase Activity from Liver and Serum

The XO activity in liver tissue and the serum was determined using the XO activity assay kit according to the manufacturer protocols (Sigma, USA). The XO activity was measured by a Spectra Max M3 microplate reader (Molecular Devices, San Jose, CA, USA) using previously published protocols [21]. The enzyme reaction was initiated by the addition of the 0.5 mM xanthine as substrate, and the concentration of uric acid was measured every minute for 5 min. The XO inhibitor allopurinol was used as a reference. The hepatic XO activity was normalized by the total protein level.

### 2.7. Kidney Histopathological Examination

Kidney tissues were removed, immediately fixed for 1 day in formalin, and then embedded in paraffin. Each specimen was cut 6 µm thick, and the sections were stained with hematoxylin and eosin reagent. The stained sections were then visualized under microscopy.

### 2.8. Western Blot Analysis

Kidney tissues were homogenized in a pro-prep extraction solution (Intron, Seoul, Korea) and then centrifuged (13,000× *g*, 4 °C) for 15 min. The supernatant was used for the Western blotting analysis of the targeted proteins. Total protein levels were determined by a DC protein assay with bovine serum albumin (Bio-Rad, Hercules, CA, USA). The primary antibodies included β-actin (Santa Cruz, Dallas, TX, USA), glucose transporter 9 (GLUT9, MyBioSource, San Diego, CA, USA), urate transporter 1 (URAT1; MyBioSource, San Diego, CA, USA), and organic anion transporter 1 (OAT1; MyBioSource, San Diego, CA, USA). The bands from the membrane were visualized by ECL detection reagent using an ImageQuant LAS 4000 (GE Healthcare Life Sciences, Seoul, Korea). The density from the obtained images was determined using Image J1.49 software of NIH (Bethesda, MD, USA). The visualized target protein levels were normalized to β-actin. ATP-binding cassette transporter G2 (ABCG2) expressions were obtained using a rat ABCG2 ELISA kit (MyBioSource) and normalized by total protein levels.

### 2.9. Measurement of Urate Uptake Using UART1-Expressing Oocytes

The hURAT1-expressing oocytes from *Xenopus* were constructed as previously described [20]. The URAT1 inhibitor benzbromarone (TCI, Tokyo, Japan) was treated as a reference. The oocytes were incubated in a reaction buffer with Dulbecco’s phosphate-buffered saline (DPBS; Sigma-Aldrich, St. Louis, MO, USA) and an ND solution (pH 7.4; 5 mM 4-(2-Hydroxyethyl)) piperazine-1-ethane-sulfonic acid buffer, 2 mM KCl, 1 mM magnesium chloride, 1.8 mM calcium chloride, and 96 mM sodium chloride), with 1 mM pyrazine carboxylic acid (Sigma-Aldrich, St. Louis, MO, USA) at 37 °C. The oocytes were further incubated in a solution containing various concentrations of EM (1, 10, and 100 μg/mL) with 50 μM [14C] uric acid (Moravek Biochemicals, Brea, CA, USA) at 37 °C. After 60 min, uric acid uptake was stopped by a supplement of cold DPBS. After stopping, the oocytes were dissolved in a solution of 0.1N sodium hydroxide and 10% sodium dodecyl sulfate, and the radioactivity was counted by liquid scintillation.

### 2.10. Statistics

All data were expressed as the means ± standard error of the mean. The significant differences were statistically evaluated by one-way analysis of variance followed by Dunnett’s test for post hoc analysis using GraphPad Prism 7.05 software. The difference was considered statistically significant when the *p* value was less than 0.05.

## 3. Results

### 3.1. Effects of Eggshell Membrane on Serum Uric Acid Secretion

A single PO injection for 1 day significantly increased serum uric acid levels in PO-injected hyperuricemic rats compared with those in normal rats (Figure 1). Meanwhile, EM at 100 and 200 mg/kg and allopurinol (positive control) effectively reduced serum uric acid in PO-injected hyperuricemic rats. Therefore, the next studies were performed to identify the dose-dependent activities and mechanisms of EM.

### 3.2. Effects of Eggshell Membrane on Serum and Urine Uric Acid Levels

A PO injection for 5 days significantly elevated the serum uric acid levels in the PO group compared with those in the normal group (Figure 2a). The treatment of EM at 100 and 200 mg/kg, as well as allopurinol and benzbromarone, significantly reduced the serum uric acid levels in the PO group. In addition, to evaluate renal function, creatinine and blood urea nitrogen (BUN) concentrations were measured. Serum creatinine concentrations were not different among the groups (Figure 2b). Serum BUN concentrations were raised in the PO group, but these levels decreased in the EM 200 mg/kg group and the positive control (allopurinol and benzbromarone) groups (Figure 2c).

To evaluate the effect of EM on uric acid excretion, urine uric acid levels were examined. The PO group had significantly decreased levels of urine uric acid, which increased by the administration of EM (100 and 200 mg/kg) and benzbromarone (Figure 2d). These data demonstrated that EM might increase kidney urate extraction to decrease serum uric acid levels in hyperuricemic rats.

### 3.3. Effects of Eggshell Membrane on Renal Inflammation

Mild tubular dilatation, vacuolar degeneration of the tubular epithelial cell, swelling, and infiltration of inflammatory cells were observed in the kidney of PO-injected hyperuricemic rats (Figure 3a). However, EM effectively improved these renal histopathological changes in hyperuricemic rats. Furthermore, hyperuricemic rats demonstrated increased serum levels of proinflammatory cytokine IL-1β (Figure 3b). The EM at all concentrations remarkably downregulated the serum levels in hyperuricemic rats. The administration of benzbromarone increased the serum IL-1β levels more than PO-injected hyperuricemic rats, although not significantly, and it is probably the side effect of this drug. These results suggest that EM effectively improved kidney inflammation and function in hyperuricemic rats.

### 3.4. Effects of Eggshell Membrane on XO Activity

The XO in the liver induces uric acid production. Thus, to define the mechanism of the inhibitory effect of EM on hyperuricemia, the XO activity in the serum and liver was evaluated. Hepatic XO activity was significantly increased in the PO group (Figure 4a,b). The XO inhibitor, allopurinol, significantly decreased XO activity in the serum and liver. However, EM at all doses did not change XO activity. These data indicate that EM does not inhibit XO activity and reduce uric acid production.

### 3.5. Effects of Eggshell Membrane on Urate Excretion

To evaluate the effect of EM on urate excretion, the expression of several renal transporters was examined. The EM at 100 and 200 ug/mL, and benzbromarone, significantly lowered URAT1protein levels in the kidney (Figure 5a,b). However, the EM did not change GLUT9 protein levels (Figure 5c). The EM at 200 ug/mL increased OAT1 protein levels, and the EM administration at all concentrations increased ABCG2 levels (Figure 5d,e).

### 3.6. Effect of EM on In Vitro URAT1 Uptake in Oocytes

In addition, the effect of EM on in vitro urate uptake system was confirmed. The EM treatment at 1, 10, and 100 ug/mL exhibited potent uric acid uptake inhibition in in vitro hURAT1-overexpressing oocytes (Figure 6). These data indicate that EM regulates the expression of urate transporters to increase urate excretion.

## 4. Discussion

In humans, the kidneys play an important role in uric acid homeostasis, as more than 70% of urate excretion is renal, while the remaining 30% is excreted in the feces from the intestine [22,23]. However, urate excretion may be impaired by renal disorders, leading to hyperuricemia. The excretion of uric acid relies on transporter proteins in the proximal tubules of the kidney to control the secretion of uric acid in the blood and filtrate [24]. Renal urate transporters URAT1, located at the luminal membrane, and GLUT9, at the apical membrane of the kidney proximal tubules, comprise the primary pathway of urate reabsorption in the kidney [25,26]. OAT1, located in the basolateral membrane of the proximal tubule, and ABCG2, located at the apical membrane, are mainly responsible for urate secretion from the blood to epithelial cells [27]. The regulation of these urate transporters is considered a major therapeutic target for hyperuricemia.

In this study, the EM effectively decreased renal URAT1 levels in PO-injected hyperuricemic rats, although GLUT9 protein expressions did not change. Moreover, OAT1 and ABCG2 protein expressions were upregulated in the kidney after the EM treatment, thereby exhibiting a uricosuric effect. This study confirmed the inhibitory activity of the EM on urate uptake by URAT1 in URAT1-overexpressing oocytes and that this effect was effective for reversing URAT1-promoted urate uptake. In total, 90% of urate reabsorption is obtained through renal URAT1 [28]. The control of these kidney urate transporters by the EM treatment may contribute to the promotion of renal uric acid excretion in hyperuricemia rats.

Creatinine and urea nitrogen are biomarkers of abnormal kidney function, indicating the ability of the kidney to excrete protein metabolites [24]. The results demonstrated that PO increases serum uric acid and BUN concentrations with renal dysfunction. However, the 200 mg/kg EM reversed this increase with improved renal function.

The enzyme XO catalyzes the oxidation of hypoxanthine or xanthine to uric acid [28]. The XO inhibitors, such as allopurinol, reduce serum uric acid concentrations and the overproduction of reactive oxygen species primarily associated with XO overactivity, which often generates inflammatory cell damage to the vascular endothelium, contributing to the development of metabolic syndrome and cardiovascular disorders [29]. Therefore, the inhibition of XO overactivation may be a therapeutic target to control the harmful effects of excess uric acid. However, our study showed that EM did not reverse serum and liver XO activity in rats with PO-induced hyperuricemia.

Hyperuricemia may be a risk element for kidney dysfunction [23]. In the present study, kidney dysfunction was characterized by increased serum BUN and inflammatory cytokine IL-1β levels and renal inflammation in hyperuricemic rats. These renal dysfunctions were attenuated by the EM treatment, suggesting an improvement in renal inflammation. However, the drug benzbromarone elevated serum IL-1β levels in hyperuricemic rats, and renal damage was reported as a primary adverse effect of uricosuric drugs, including benzbromarone and probenecid [30].

EM contains naturally occurring glucosamine, chondroitin sulfate, hyaluronic acid, collagen, peptides, and other sulfur-containing amino acids [8]. Previous studies reported that glucosamine and chondroitin have demonstrated anti-inflammatory and anti-osteoarthritic effects [31,32]. Hyaluronic acid exhibited anti-inflammatory and antihyperuricemic effects in monosodium urate crystal-induced animal models for acute gouty arthritis and hyperuricemia [33]. The results of these studies show the potential usefulness of these components to prevent and treat hyperuricemia. However, our study did not show which components of the EM enhanced the excretion of uric acid. The effects of bioactive components derived from EM must be further investigated.

In this study, we demonstrated that the PO-induced hyperuricemia and kidney inflammation was inhibited by EM treatment. Therefore, EM could be a promising antihyperuricemic agent. Suppose an effective dose of EM in rats is 100 mg/kg (maximum dose; 200 mg/kg). Based on body surface area, the human equivalent dose of EM is 16.2 mg/kg (972 mg/day in adult). From these in vivo studies, the recommended daily dosage of EM in mild conditions is 486 mg or 972 mg daily, and a maximum dose is 1944 mg daily for severe conditions. However, the dosage can differ depending on the treatment duration and age. Based on these results, a human clinical study of EM will proceed in the near future.

## 5. Conclusions

These findings demonstrate for the first time that EM improves hyperuricemia by promoting renal uric acid excretion. These effects are achieved by regulating urate transporters and promoting urate excretion.

## Figures and Tables

**Figure 1 nutrients-13-03323-f001:**
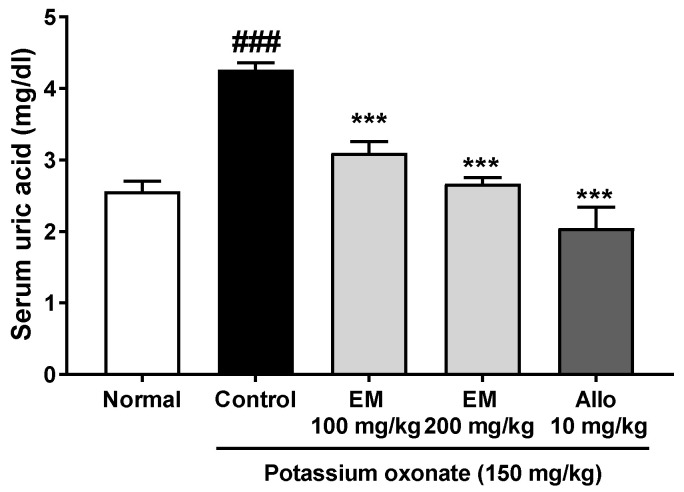
The effects of eggshell membrane (EM) on serum uric acid in hyperuricemic rats (1-day treatment duration) (*n* = 5 per group). Allo, allopurinol. ### *p* < 0.001 vs. the normal group; *** *p* < 0.001 vs. the potassium oxonate group.

**Figure 2 nutrients-13-03323-f002:**
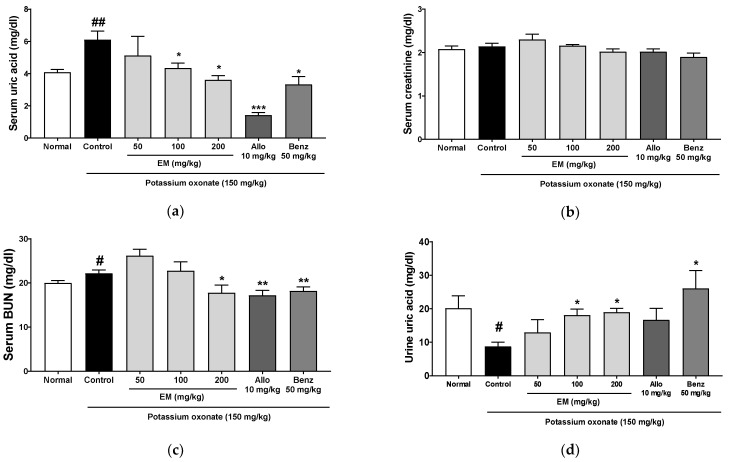
The effects of EM on the parameters of serum uric acid levels and kidney function in rats with potassium oxonate (PO)-induced hyperuricemia (5-day treatment duration). (**a**) Serum uric acid. (**b**) Serum creatinine. (**c**) Serum blood urea nitrogen (BUN). (**d**) Urine uric acid; (*n* = 5 per group). Allo, allopurinol; Benz, benzbromarone. # *p* < 0.05 and ## *p* < 0.01 vs. the normal group; * *p* < 0.05, ** *p* < 0.01 and *** *p* < 0.001 vs. the PO group.

**Figure 3 nutrients-13-03323-f003:**
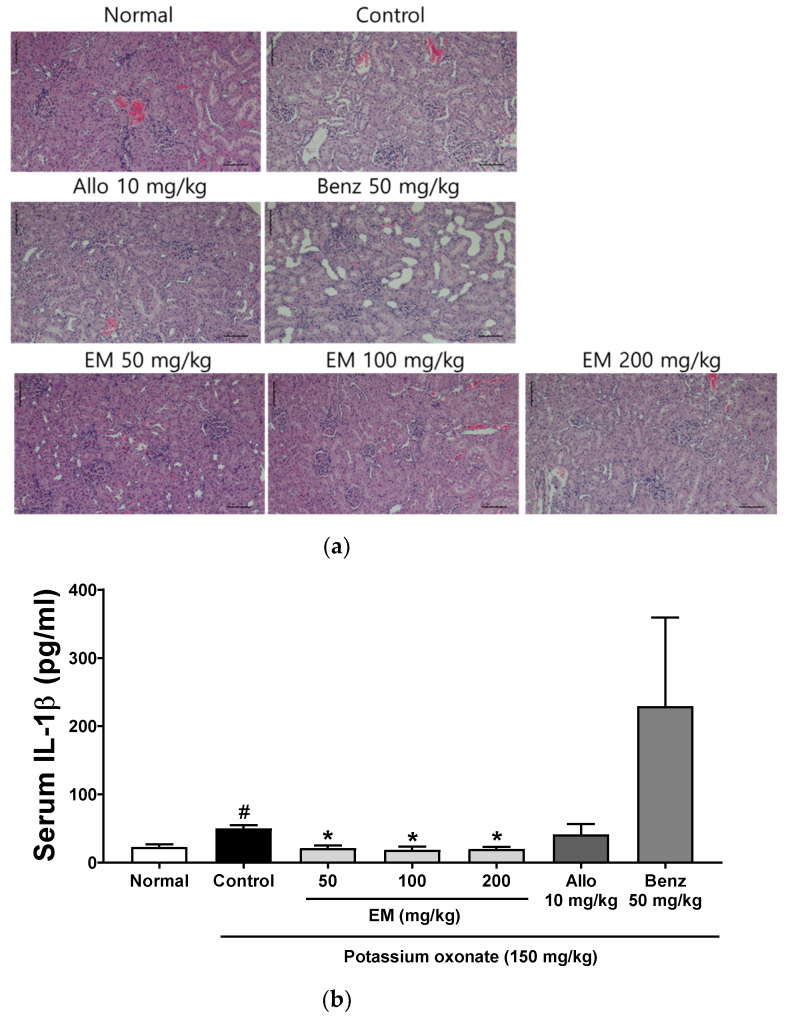
The effects of EM on renal inflammation in rats with PO-induced hyperuricemia: (**a**) Renal histopathological changes (original magnification, 200×). (**b**) Serum IL-1β levels by ELISA. (*n* = 5 per group). Allo, allopurinol; Benz, benzbromarone. # *p* < 0.05 vs. the normal group; * *p* < 0.05 vs. the PO group.

**Figure 4 nutrients-13-03323-f004:**
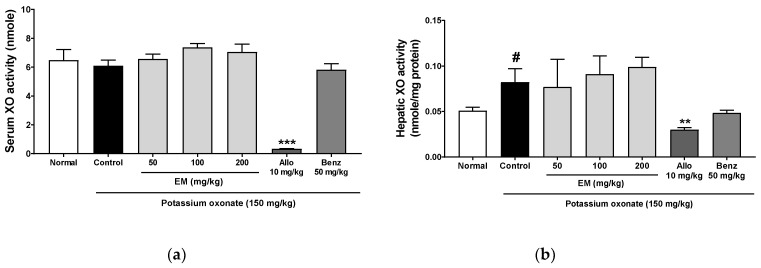
The effects of EM on xanthine oxidase activity in rats with PO-induced hyperuricemia. (**a**) Serum XO activity. (**b**) Hepatic XO activity; (*n* = 5 per group). Allo, allopurinol; Benz, benzbromarone. # *p* < 0.05 vs. the normal group; ** *p* < 0.01 and *** *p* < 0.001 vs. the PO group.

**Figure 5 nutrients-13-03323-f005:**
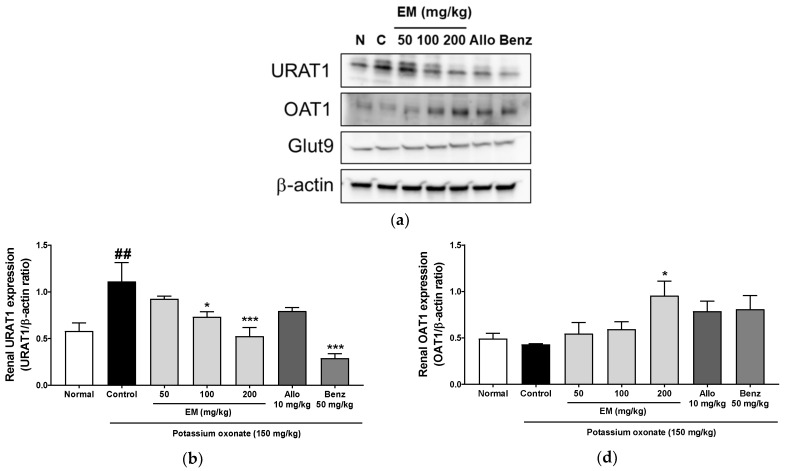
The effects of EM on the expression of urate transporters in rats with PO-induced hyperuricemia. (**a**) Renal protein expression levels of URAT1, GLUT9, and OAT1 urate transporters by Western blot analysis. Protein expression ratio of (**b**) URAT1, (**c**) GLUT9, and (**d**) OAT1. (**e**) ABCG2 levels by ELISA; (*n* = 5 per group). Allo, allopurinol; Benz, benzbromarone. ## *p* < 0.01 vs. the normal (N) group; * *p* < 0.05 and *** *p* < 0.001 vs. the PO group.

**Figure 6 nutrients-13-03323-f006:**
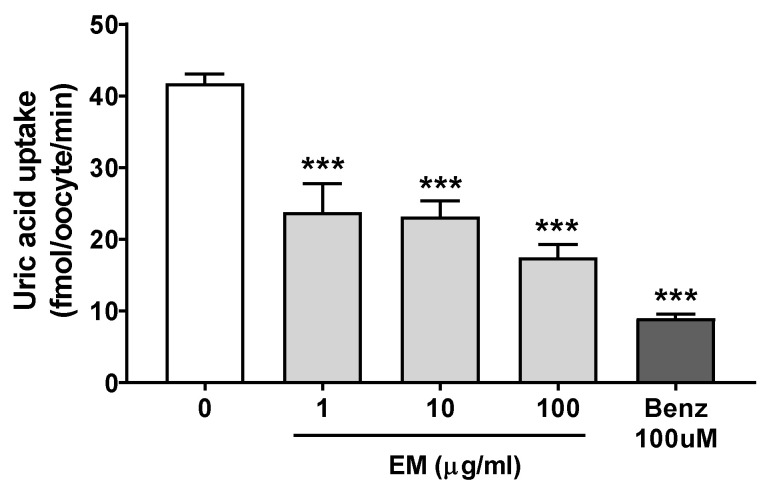
The effect of EM on uric acid uptake in in vitro hURAT1-overexpressing oocytes; (*n* = 5 per group). Benz, benzbromarone. *** *p* < 0.001 vs. the no EM-treated vehicle group.

## Data Availability

The data presented in this study are available in this article.

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
