# Peer review of "Eggshell Membrane Ameliorates Hyperuricemia by Increasing Urate Excretion in Potassium Oxonate-Injected Rats"

_nutrients, 2021, doi:10.3390/nu13103323_

Round 1
Reviewer 1 Report
The authors investigated the effects of eggshell membrane on markers of hyperuricemia in potassium oxonate-injected rats. The manuscript is generally well-written and has some good information for readers. The reviewer has a few comments and recommend major revisions.
Specific points:
Line 79: What’s the form of eggshell membrane? Is it fresh, dried or in form of supplement?
Lines 79-96: It seems that the Experiment 2 has already covered the different treatments of Experiment 1. Why the authors designed two experiments? Why designed the 5-day treatment duration for Experiment 2 but 1-day treatment for Experiment 1?
Lines 88 and 95: How many times did you treat the rats with PO injection and then oral treatment in a day? Did the rats receive the PO injection and oral treatment everyday during the 5 days in Experiment 2?
What is the human equivalent dose of EM?
Figures 1 and 2: please clarify the treatment durations in the figure legend, since the two experiments have different treatment days.
Line 182: repeated “**p < 0.01”. Please annotate triple asterisk.
Figure 3: Was there any significant difference between Control and Benz groups?
Line Figure 5: Figures 5(b) and 5(c) are not shown in the manuscript.
It may be worthwhile to discuss why EM is effective. Does it have any bioactive components related to the health benefit?
Please discuss the limitations of this study.
Author Response
- Line 79: What’s the form of eggshell membrane? Is it fresh, dried or in form of supplement?
Response: Eggshell membrane was supplemented as dried power form (line 79).
- Lines 79-96: It seems that the Experiment 2 has already covered the different treatments of Experiment 1. Why the authors designed two experiments? Why designed the 5-day treatment duration for Experiment 2 but 1-day treatment for Experiment 1?
Response: Experiment 1 is a preliminary acute study for investigate anti-hyperuricemic effect of eggshell membrane. To study the dose-dependent effects and mechanisms of eggshell membrane on urate production and excretion, the 5-day treatment of Experiment 2 was performed. The long-term treatment is need to investigate the effects, toxicity, and molecular mechanisms on urine and tissues in addition to blood (line 85 and 90).
- Lines 88 and 95: How many times did you treat the rats with PO injection and then oral treatment in a day? Did the rats receive the PO injection and oral treatment everyday during the 5 days in Experiment 2?
Response: In Experiment 1, PO injection and EM oral treatment was treated once in a day. In Experiment 2, PO injection and EM oral treatment was treated everyday during the 5 days (line 95).
- What is the human equivalent dose of EM?
Response: Suppose effective dose of EM in rat is 100 mg/kg. Based on body surface area, the human equivalent dose of EM is 16. 2 mg/kg (972 mg/day).
- Figures 1 and 2: please clarify the treatment durations in the figure legend, since the two experiments have different treatment days.
Response: I added the treatment durations in the figure legend 1 and 2 according to the reviewer’s comment (line 163 and 180).
- Line 182: repeated “**p < 0.01”. Please annotate triple asterisk.
Response: I changed to ***p < 0.001 (line 182).
- Figure 3: Was there any significant difference between Control and Benz groups?
Response: There was no significant difference between the two groups.
- Line Figure 5: Figures 5(b) and 5(c) are not shown in the manuscript.
Response: I revised the legend of Figure 5 (line 219).
- It may be worthwhile to discuss why EM is effective. Does it have any bioactive components related to the health benefit?
Response: EM contains naturally occurring glucosamine, chondroitin sulfate, hyaluronic acid, collagen, peptides and other sulphur-containing amino acids [8]. Previous studies re-ported that glucosamine and chondroitin of EM has demonstrated anti-inflammatory and anti-osteoarthritic effects [31-32]. Hyaluronic acid exhibited an anti-inflammatory and antihyperuricemic effects in monosodium urate crystal-induced animal models for acute gouty arthritis and hyperuricemia [33]. The results of these studies show the potential usefulness of these components to prevent and treat hyperuricemia. I added these sentence (line 268-273).
10.Please discuss the limitations of this study.
Response: Our study did not show which components of EM enhanced the excretion of uric acid. The effects of bioactive components derived from EM must be further investigated. I added the sentence (line 273-276).

Reviewer 2 Report
The title of this paper is "Eggshell membrane ameliorates hyperuricemia by increasing urate excretion in potassium oxonate–injected rats". The results of their feeding study on potassium oxonate–injected rats proved that Eggshell membrane (EM) reduces serum uric acid levels and increases urine uric acid levels in hyperuricemia rats.. It down-regulates renal renal urate transporter 1 (URAT1) protein levels and up-regulates organic anion transporter 1 (OAT1) and ATP binding cassette transporter G2 (ABCG2), but does not change the expression of glucose transporter 9 (GLUT9). EM does not change the xanthine oxidase activity in the liver and serum. EM significantly reduces kidney inflammation and serum interleukin 1β levels. Their related experimental design and results of potassium oxonate–injected rats fed EM support the research conclusion that EM can exhibit anti-hyperuricemia effects by promoting renal urate excretion and regulating renal urate transporter. The research results have nutritional therapy and academic reference value.
It is recommended to strengthen the discussion in the discussion of how much EM dosage is required for human intake to be an effective and safe supplement to achieve anti-hyperuricemia effect.
Author Response
- It is recommended to strengthen the discussion in the discussion of how much EM dosage is required for human intake to be an effective and safe supplement to achieve anti-hyperuricemia effect.
Response: Suppose effective dose of EM in rat is 100 mg/kg. Based on body surface area, the human equivalent dose of EM is 16. 2 mg/kg (972 mg/day). From these in vivo studies, the recommended daily dosage of EM in mild condition is 486 mg or 972 mg daily and a maximum dose is 1,944 mg daily for severe condition. However, the dosage can be different depending on the treatment duration and age. Based on these results, we will be proceeding with a human clinical study of EM in the near future (line 279-283).

Round 2
Reviewer 1 Report
The Figures look good in this version. I have no more question. I think the manuscript is acceptable for publication.